# Internal Resistor Effect of Multilayer-Structured Synaptic Device for Low-Power Operation

Hyejin Kim [1], Geonhui Han [1], Seojin Cho [1], Jiyong Woo [2] and Daeseok Lee [1],*

[1] Department of Electronic Materials Engineering, Kwangwoon University, Seoul 01897, Republic of Korea
[2] School of Electronic and Electrical Engineering, Kyungpook National University, Daegu 41566,
Republic of Korea; jiyong.woo@knu.ac.kr
* Correspondence: leeds@kw.ac.kr

**Abstract:** A synaptic device with a multilayer structure is proposed to reduce the operating power of neuromorphic computing systems while maintaining a high-density integration. A simple metal-insulator-metal (MIM)-structured multilayer synaptic device is developed using an 8-inch wafer-based and complementary metal-oxide-semiconductor (CMOS) fabrication process. The three types of MIM-structured synaptic devices are compared to assess their effects on reducing the operating power. The obtained results exhibited low-power operation owing to the inserted layers acting as an internal resistor. The modulated operational conductance level and simple MIM structure demonstrate the feasibility of implementing both low-power operation and high-density integration in multilayer synaptic devices.

**Keywords:** CMOS compatibility; MIM structure; multilayer synaptic device; low-power operation; inner resistor effect

## 1. Introduction

The recent exponential growth in unstructured data has led to a significant increase in the amount of data required for efficient processing [1,2]. However, conventional von Neumann computing systems have limitations that result in slow data processing owing to the bottleneck effect caused by the sequential transfer of data between the central processing unit and memory [3–5]. To address this issue, researchers have explored neuromorphic computing systems that use parallel data processing, which enables faster and more energy-efficient processing of large amounts of data [6–8]. To implement this neuromorphic computing system in the hardware, current-based vector–matrix multiplication (VMM) is commonly used via a synaptic device array [9–11]. Because a larger synaptic device array can process more data in parallel, the high-density integration of the synaptic device is necessary. For this purpose, in this research, a simple two-terminal (2T)-based metal–insulator–metal (MIM)-structured memristor which has been studied for memory application is utilized as the synaptic device [12–18].

The 2T-based memristor devices have been investigated, including resistive random-access memory (ReRAM), phase-change memory (PCM) [19], ferroelectric random-access memory (FeRAM) [20,21], and Magnetic random-access memory (MRAM) [22]. Among these memristor devices, ReRAM is the most attractive candidate owing to its simple structure, high-density integration, fast switching speed, and excellent scalability [23–28]. Although the memristor-based synaptic device array can lead to faster parallel data processing using VMM, further research is required to minimize its power consumption. However, ReRAM has been studied for memory application [16,29], research on the device operation mechanism [29–31], and research on ReRAMs composed of materials that are not CMOS-compatible [32,33]. Thus, in this study, a memristor-based 2T synaptic device with a multilayer structure was proposed to reduce the operating power while maintaining high-density integration. Moreover, 8 inch wafer-based CMOS fabrication processes and an

oxide-based $W/TaO_X/AlO_X/WO_X/TiN$ stack were used to assess the feasibility of mass production. The obtained result showed that the $AlO_X$ layer acted as an internal resistor (and barrier layer) without degradation of the synaptic characteristics and exhibited a low-power operation.

## 2. Materials and Methods

A simple MIM-structured memristor was fabricated to realize the high-density integration of multilayer synaptic devices, as shown in Figure 1. The three types of devices were fabricated to evaluate their effects on reducing the operating power. The $W/WO_X/TiN$, $W/TaO_X/WO_X/TiN$, and $W/TaO_X/AlO_X/WO_X/TiN$ stacks were named the single layer, double layer, and triple layer, respectively.

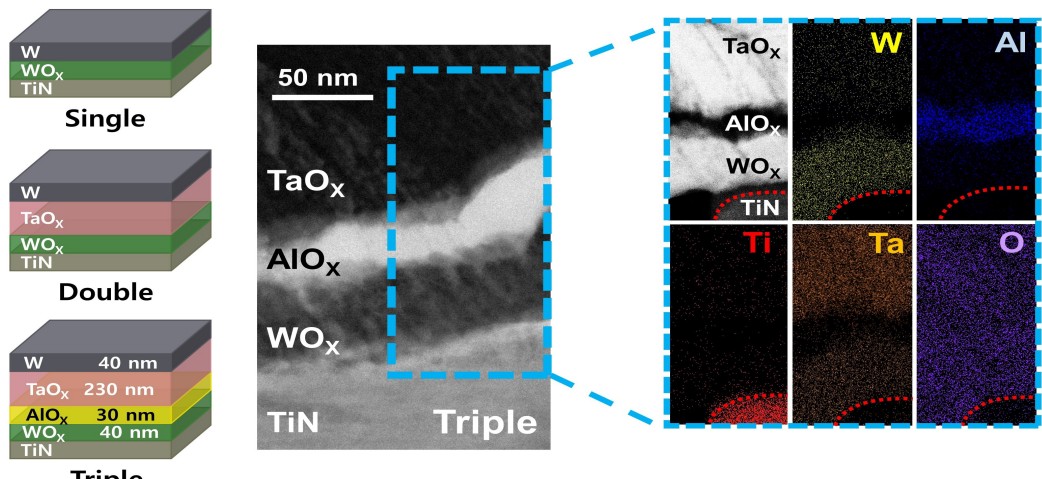

**Figure 1.** Schematic diagrams of three types of devices with MIM structures; single layer, double layer, and triple layer. Cross-sectional transmission electron microscopy and energy dispersive spectrometry mapping image of the triple layer.

First, a photolithography process was performed to pattern the device. For photolithography, AZ 5214E photoresist (AZ Electronic Materials, Bridgewater, NJ, USA) was applied to the entire wafer using a spin coater. Then, the AZ 300 MIF developer was used for development. A $WO_X$ layer was deposited on the TiN bottom electrode using a typical radio frequency (RF) sputtering system, and a W layer was formed as the top electrode (TE) (called the single layer). Multilayer structures, such as $W/TaO_X/WO_X/TiN$ (called the double layer) and $W/TaO_X/AlO_X/WO_X/TiN$ stack (called the triple layer), were developed and compared to assess their effects on reducing the operating power of the device. All layers were deposited using a sputtering system, and the deposition parameters of each layer are as follows. A 40 nm $WO_X$ channel was deposited by reactive sputtering using a $WO_3$ target in a 4:1 ratio of Ar and $O_2$ mixed ambient gas. Then, a 30 nm thick $AlO_X$ layer and a 230 nm thick $TaO_X$ layer were deposited using an $Al_2O_3$ and $Ta_2O_5$ target in Ar as the ambient gas. Finally, a 50 nm thick W layer was deposited as the top electrode in ambient Ar gas. $WO_X$ and $AlO_X$ were deposited at a working pressure of 5 mTorr, while $TaO_X$ and W were deposited at 10 m Torr.

Figure 1 shows a cross-sectional transmission electron microscopy (TEM) and energy dispersive spectrometry (EDS) mapping image of the fabricated triple layer. The characteristic X-ray energy of Ta and W elements is 1.709 and 1.774 keV, respectively [34]. Therefore, the W element in the $TaO_X$ region and the Ta element in the $WO_X$ region may overlap. The fabrication processes were based on 8 inch wafer-based CMOS fabrication processes; more details are described in reference [35]. All electrical measurements were conducted using a semiconductor parameter analyzer (HP 4156A) and a pulse generator (Agilent 81110A).

## 3. Results and Discussion

As mentioned above, the synaptic devices of single, double, and triple layers were fabricated. To confirm the synaptic characteristics of each device, each weight-update curve was measured (Figure 2a–c). The inset of Figure 2a–c show the pulse conditions for potentiation (conductance increase) and depression (conductance decrease). In the single layer, it exhibited resistive switching, which refers to resistance changes from a high-resistance state to a low-resistance state in the negative bias region, and vice versa. When a positive bias is applied to the TE, the oxygen ions of the $WO_X$ layer are migrated to the TE. This migration results in the formation of an induced oxide layer at the interface between the $WO_X$ layer and TE, resulting in decreased conductance. The thickness of the induced oxide layer increased as a continuous positive pulse bias was applied, and thus the conductance was modulated (Figure 2a,d) [36,37]. In contrast, the weight update curve occurs at the opposite polarity for the double and triple layer (Figure 2b,c). The inset of Figure 2b shows the current-voltage (I–V) curve characteristic of the double layer. Gradual resistive switching of the double layer was observed under optimized conditions. The set process in the positive bias and the reset process in the negative bias are observed. Switching behavior occurred in the $WO_X$ layer depending on the mobile oxygen ions between $WO_X$ and $TaO_X$ layers [35,38]. When the positive bias was applied to the TE, the oxygen ions in the $WO_X$ layer moved to the $TaO_X$ layer. Thus, the amount of oxygen vacancies in $WO_X$ increased, resulting in the potentiation process. Conversely, when the negative bias was applied, the oxygen ions that had moved to the $TaO_X$ layer during the potentiation process moved back to the $WO_X$ layer, resulting in the depression process.

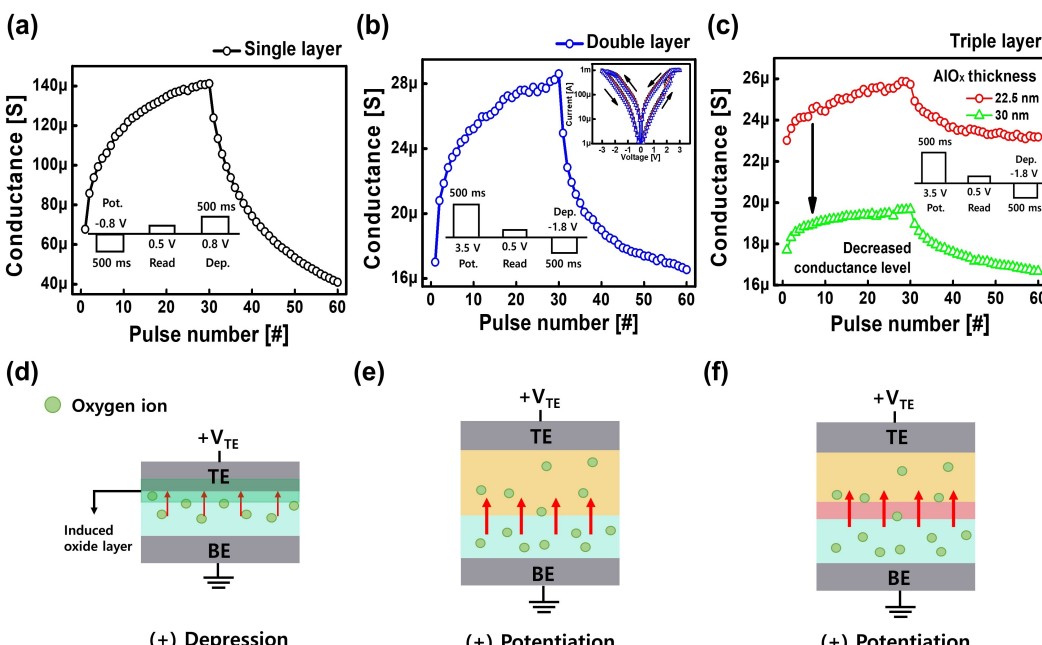

**Figure 2.** Synaptic characteristics of the weight-update curve in the (**a**) single, (**b**) double, and (**c**) triple layer. The inset shows the optimized pulse amplitude and width (Pot.: −0.8 V, 500 ms/Dep: +0.8 V, 500 ms for single layer and Pot: +3.5 V, 500 ms/Dep: −1.8 V, 500 ms for double and triple layer). (**c**) Potentiation and depression depend on the thickness of the $AlO_X$ layer in the triple layer. (**d**–**f**) Schematic diagram of the operation mechanism in the single, double, and triple layer, respectively.

To achieve synaptic characteristics based on this operating mechanism, the fabrication conditions (such as the Ar: $O_2$ ratio of the $WO_X$ layer and the working pressure of the $TaO_X$) were optimized, as shown in Figure 3. A higher initial resistance was observed during the deposition of the $WO_X$ when the Ar: $O_2$ ratio was increased (Figure 3a). However, resistive switching was only obtained when the ratio of Ar to $O_2$ was 20:5. This result can be explained in terms of the oxygen vacancy density in the $WO_X$ layer [39] (Figure 3b).

When the $Ar:O_2$ ratio changed to 20:1, more oxygen vacancies were present in the $WO_X$ layer, resulting in an electrically short state. In contrast, when the $Ar:O_2$ ratio was 20:10, sufficient oxygen ions were supplied during the deposition of the $WO_X$. Consequently, an insulating $WO_X$ layer was formed, leading to an electrically insulating behavior.

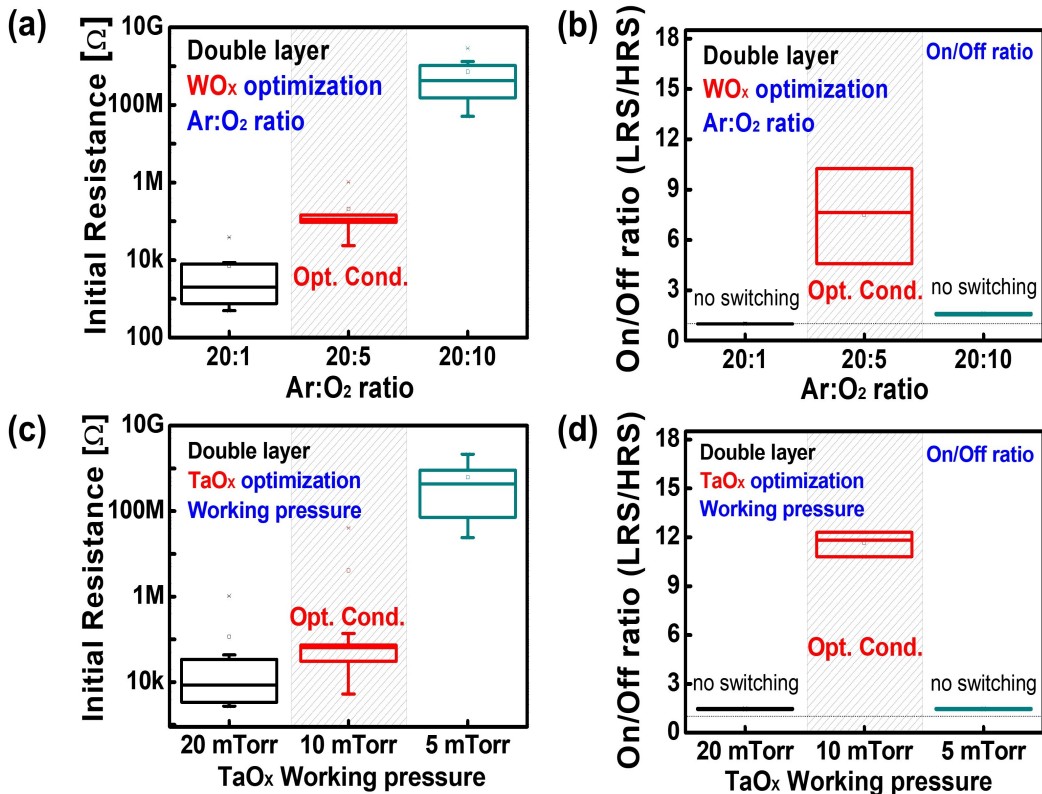

**Figure 3.** (**a**) Initial resistance and (**b**) on/off ratio depending on the $Ar:O_2$ ratio of the $WO_X$ layer. The resistive switching characteristic appears only under the 20:5 optimized condition. (**c**) Initial resistances with varying $TaO_X$ layer working pressures. (**d**) On/off ratio according to the $TaO_X$ layer working pressures. The resistive switching characteristic appears only under 10 mTorr. (Reproduced from Ref. [35] with permission from the Royal Society of Chemistry.)

Based on the optimized $WO_X$ oxygen partial pressure condition, the working pressure of the $TaO_X$ layer was also varied to achieve synaptic characteristics, as shown in Figure 3c,d. When the working pressure was changed from 20 to 10 and 5 mTorr, resistive switching was observed only at 10 mTorr. Considering that a higher working pressure can result in a porous film, deposition at 20 mTorr forms a more porous $TaO_X$ layer [37,40]. Similarly, a denser $TaO_X$ layer was deposited at 5 mTorr. Because the effective area of the interface between the $TaO_X$ and $WO_X$ layers can be increased by higher porosity, more oxygen absorption, resulting in an electrically short state, can occur at 20 mTorr. Additionally, at 5 mTorr, the reduced effective interfacial area and formation of a denser $TaO_X$ layer prevented oxygen absorption. Based on these results, conditions such as an $Ar:O_2$ ratio of 20:5 and a working pressure of 10 mTorr were selected as the optimal fabrication conditions for the $WO_X$ and $TaO_X$ layers.

The double layer exhibited a lower conductance level than the single layer; however, it was still unacceptably high for the low-power operation of synaptic devices. To further reduce the operating power of the synaptic device, an $AlO_X$ layer was inserted into the interface between the $TaO_X$ and $WO_X$ layers (triple layer). The $AlO_X$ layer was added between the $TaO_X$ and $WO_X$ layers, rather than elsewhere, to obtain the synaptic characteristic. When the $AlO_X$ layer was added to the interface between the $WO_X$ and TiN layers ($W/TaO_X/WO_X/AlO_X/TiN$), no switching characteristic was observed. The triple layer

has an operating mechanism similar to the double layer. The switching occurs in the $WO_X$ layer according to the mobility of oxygen ions between the $WO_X$ layer and $TaO_X$ layer, as shown in Figure 2f. When the positive bias is applied to the top electrode, oxygen ions in the $WO_X$ layer migrate through the $AlO_X$ layer to the $TaO_X$ layer, causing switching in the $WO_X$ layer. Thus, the potentiation process occurs in which the conductance increases under a positive bias. The conductance level of potentiation and depression decreased with the insertion of the $AlO_X$ layer. The thickness of the $AlO_X$ layer was varied from 22.5 to 30 nm for optimization. The initial resistance increased with increasing $AlO_X$ layer thickness. Owing to the increased initial resistance, the conductance levels of potentiation and depression decreased.

The conductance levels of potentiation and depression were compared in three types of synapse devices. The conductance levels of potentiation and depression decreased with increasing number of layers (Figure 4a). The synaptic characteristics of the devices were verified by normalizing and comparing the potentiation and depression behaviors of the single, double, and triple layers using Equation (1), where $G_{max}$ and $G_{min}$ are the maximum conductance state and minimum conductance state, respectively. The normalized synaptic potentiation and depression behaviors of each device were similar, indicating that the multilayer structure can reduce the operating power without significantly degrading the synaptic characteristics (Figure 4b).

$$G_{normal} = \frac{(G - G_{min})}{(G_{max} - G_{min})} \tag{1}$$

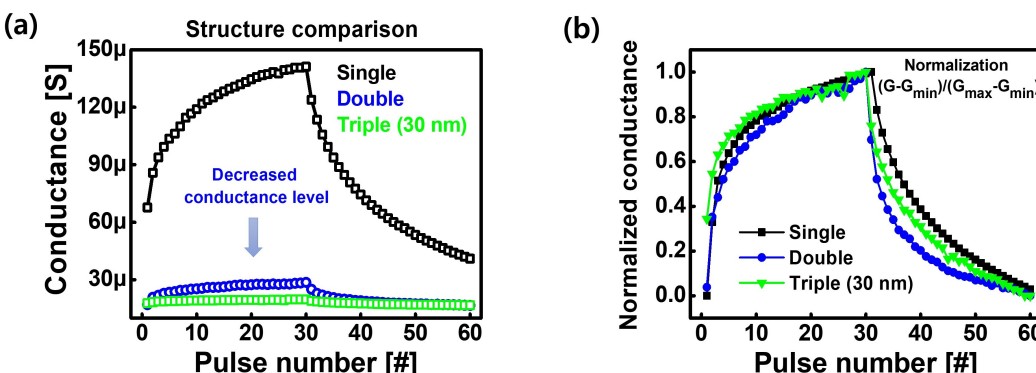

**Figure 4.** (**a**) Comparison of the conductance levels of potentiation and depression in the three types of devices. The conductance levels of potentiation and depression decreased with increasing number of layers. (**b**) Normalized conductance of the single, double, and triple layer devices in potentiation and depression curves. The plot is employed to compare the synaptic characteristics of the devices.

To investigate the role of the inserted $AlO_X$ layer, three cases, namely a double layer, a double layer with an external commercial resistor (200 k$\Omega$), and a triple layer, were compared in Figure 5. Figure 5a compares the double and triple layers, revealing an obvious decrease in the conductance level of the triple layer. As shown in Figure 5b, the conductance of double layer with an external commercial resistor was measured by connecting a 200 k$\Omega$ commercial resistor in series through the wiring outside of the double layer device. When the external resistor was connected to the double layer, the conductance level decreased. Compared with the double layer, as shown in Figure 5c, both the triple and double layers with an external resistor exhibited significantly decreased conductance levels. Furthermore, the triple layer exhibited the same operating conductance level as the double layer connected to the external resistor. This result implies that the inserted $AlO_X$ layer can serve as an internal 200 k$\Omega$ resistor to efficiently reduce the conductance level.

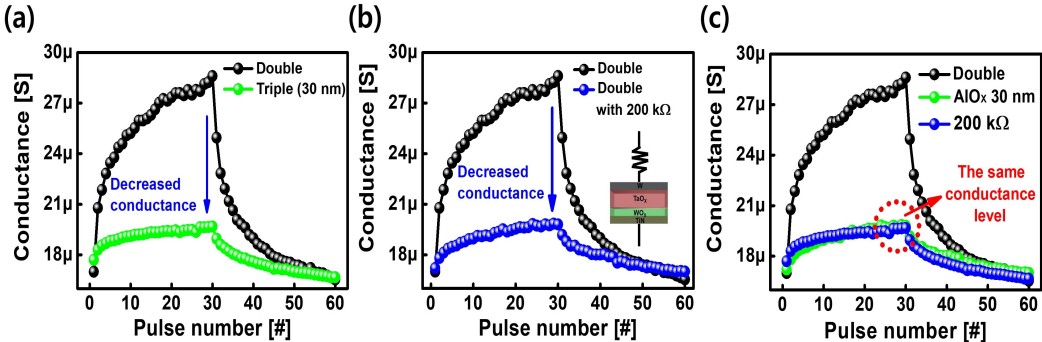

**Figure 5.** Comparison of the potentiation and depression characteristics of the double layer device with those of (**a**) a 30 nm-thick AlO$_X$ layer in the triple layer (Triple-30), (**b**) a double layer with a 200 kΩ resistor (Double-200 kΩ), and (**c**) both Triple-30 and Double-200 kΩ. The results indicate that the inserted AlO$_X$ layer can serve as an internal 200 kΩ resistor.

In addition, the composition ratio of the WO$_X$ layer, which is a switching layer, was changed compared to the double layer because the AlO$_X$ layer was inserted between the WO$_X$ layer and the TaO$_X$ layer in the triple layer. When the AlO$_X$ layer, which acts as a barrier layer (or shielding layer) [41], was deposited on the WO$_X$ layer, the amount of oxygen ions absorbed from the WO$_X$ layer was reduced. Accordingly, compared with the double layer, the oxygen vacancy density of the WO$_X$ layer of the triple layer decreases. These results were quantitatively analyzed by X-ray photoelectron spectroscopy (XPS) measurements in Figure 6. Figure 6a,b show the XPS analysis spectra of O 1s in the WO$_X$ layer of the double layer and the triple layer, respectively.

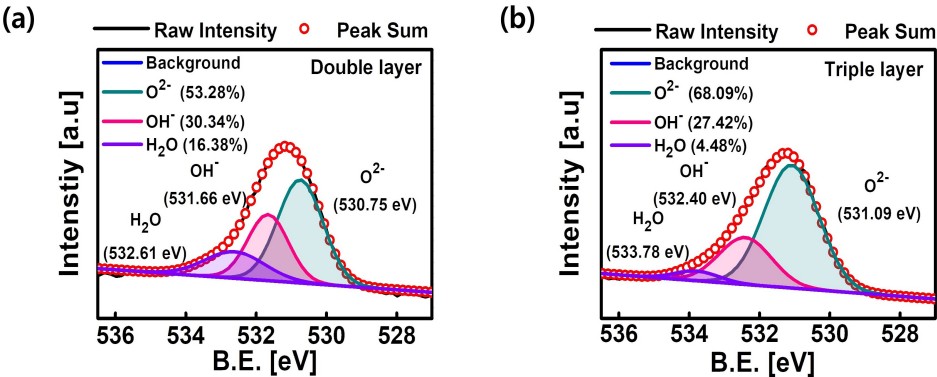

**Figure 6.** XPS analysis spectra of O 1s in the WO$_X$ layer of (**a**) the double layer and (**b**) the triple layer.

The XPS spectrum showed a broad peak, which can be deconvoluted into three individual peaks: the W-O bond peak, oxygen vacancy density, and chemisorbed oxygen species. The green peak of the double layer (530.75 eV) and the triple layer (531.09 eV) can be assigned to the oxygen atoms (O$^{2-}$) which form W-O bonds. In addition, the violet peaks represent chemisorbed oxygen species (H$_2$O). Finally, the pink peaks can be assigned to species adsorbed on the surface (OH$^-$, O$^-$, or oxygen vacancies); the OH$^-$ groups bond with the metal cations to maintain a charge balance. This implies that the intensity of the OH$^-$ peak indicates oxygen vacancy density [42,43]. The oxygen vacancy density of the double and triple layer are 30.34% and 27.42%, respectively. Therefore, the triple layer has a lower oxygen vacancy density than the double layer. The stoichiometric ratio between tungsten and oxygen can be determined from the composition ratio. In double layer, the tungsten atomic ratio is 30.76% and the oxygen atomic ratio is 69.24%. Thus, the ratio of the tungsten to the oxygen is about 1:2.25 (WO$_{2.25}$). In the same way, the atomic ratio of tungsten in the triple layer is 29.08% and the atomic ratio of oxygen is 70.92%, so the ratio is 1:2.44 (WO$_{2.44}$) (Table 1). This indicates that the WO$_X$ of the triple layer contains a smaller number of oxygen vacancies compared to the WO$_X$ of the double layer. As a result, the

$AlO_X$ layer plays the role of 200 k$\Omega$ because the defect in the switching layer ($WO_X$ layer) decreases the resistance of the $AlO_X$ layer itself. Therefore, the conductance level of the triple layer decreases.

**Table 1.** Summary of the atomic ratios of W and O of the $WO_X$ layer of the double and triple layers.

| Device | Material | Atomic Ratio (%) | Condition |
|---|---|---|---|
| Double layer $WO_{2.25}$ | W 4f | 30.76 | $WO_3$ target Ar:$O_2$ = 20:5 |
| | O 1s | 69.24 | |
| Triple layer $WO_{2.44}$ | W 4f | 29.08 | |
| | O 1s | 70.92 | |

Owing to the decreased or modulated conductance level, the synaptic device for the neuromorphic system can achieve low power consumption. The power consumption of the single, double, and triple layer was numerically calculated as shown in Figure 7a. When comparing the single and double layer, the power consumption of the double layer was slightly decreased, from 28.24 μJ to 25.03 μJ. This is because a voltage drop occurred by inserting a $TaO_X$ layer. Thus, a larger pulse amplitude is required for the double layer, and the power consumption was only slightly decreased. However, the power consumption of the triple layer was reduced by 31.2% compared to the double layer (from 25.03 μJ to 17.22 μJ), with the same pulse width and amplitude. Considering the huge size of the synaptic array in the neuromorphic system, a significant reduction in power consumption can be expected.

Additionally, to verify the influence at the system level, an image recognition simulation consisting of four-layer neural networks was conducted, as shown in Figure 7b–e. The IBM Analog Hardware Acceleration Kit (AIHWKIT)), which can simulate devices in real-world applications, is used to simulate training and inference [44]. This provides several device models. We used a "LinearStepDevice" among them. Each parameter required for the simulation was extracted from the measured potentiation/depression weight update curve of the single, double, and triple layer. The neural network was constructed with an input layer of 784 neuron nodes, hidden layer 1 of 256 neuron nodes, hidden layer 2 of 128 neuron nodes, and an output layer of 10 neuron nodes (Figure 7b). A synapse device model was used to connect each neuron node. For the implementation of the deep neural network of Figure 7b at the device level, a synaptic device acting as a weight value can be constructed by a cross-point array [45]. To perform the Multiply and Accumulation operation, the input voltage bias is applied to all row lines, and the output is obtained as a summed current by multiplying the conductance stored at the synaptic devices (Figure 7c). We utilized the Modified National Institute of Standards and Technology (MNIST) dataset (28 × 28) as an input image. Figure 7d,e show the recognition rate according to the training epoch. The image recognition rates are 85.10%, 71.51%, and 84.11% for a single, double, and triple layer when the numerically ideal case is 92.57%. Even though the triple layer has the lowest power consumption, it exhibited a similar recognition rate to others. This is because the linearity of the weight update curve was not degraded with the addition of the layer compared to the single layer. The image recognition rate reaches about 85%, which is respectable but could be even higher with a wider dynamic range.

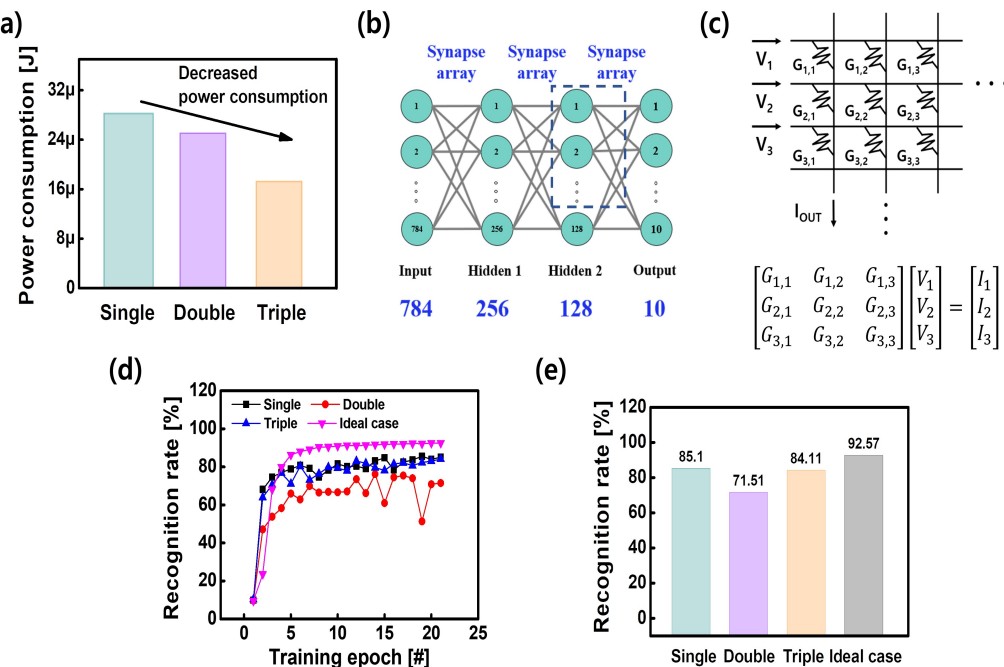

**Figure 7.** (**a**) The power consumption of the single, double, and triple layer for the MNIST pattern recognition. (**b**) Schematic of the neural network; the neural network was constructed with 784 inputs × 256 first hidden × 128 second hidden × 10 output neurons. (**c**) The crossbar array consists of vertical rows and columns with resistive synaptic devices sandwiched at each cross-point. Recognition rate (**d**) during and (**e**) after 20 epochs for single, double, triple layer, and ideal case.

## 4. Conclusions

In this study, the synaptic device with multilayer MIM-structured synaptic devices suitable for high-density integration and low-power operation were developed using 8 inch wafer-based CMOS fabrication processes. Compared to the double layer, the triple layer demonstrated a low-power operation as the power consumption was reduced by approximately 31%. The synaptic device for neuromorphic systems achieved a low-power consumption due to the reduced or modulated conductance level, because the AlO$_X$ layer inserted in the triple layer not only acts as a barrier layer but also acts as an internal resistor. In addition, the triple layer does not degrade the synaptic characteristics even when the AlO$_X$ layer is added, so the recognition rate shows the undegraded performance of 84.11%. Therefore, the obtained results demonstrate the feasibility of achieving both a low-power operation and high-density integration in multilayer synaptic devices.

**Author Contributions:** D.L. conceived and directed the research. H.K. and G.H. conducted the experiment. H.K., S.C. and G.H. analyzed the results. H.K. wrote the manuscript. J.W. revised the manuscript. All authors have read and agreed to the published version of the manuscript.

**Funding:** This research was supported by the MSIT (Ministry of Science and ICT), Korea, under the ITRC (Information Technology Research Center) support program (IITP-2023- RS-2022-00156225) supervised by the IITP (Institute for Information & Communications Technology Planning & Evaluation), the Research Grant of Kwangwoon University in 2023, and the National Research Foundation of Korea (NRF) grant funded by the Korea government (MIST) (2009-0082580).

**Data Availability Statement:** The data presented in this study are available on request from the corresponding author.

**Conflicts of Interest:** The authors declare no conflicts of interest.

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
