# Peer review of "Internal Resistor Effect of Multilayer-Structured Synaptic Device for Low-Power Operation"

_nanomaterials, doi:10.3390/nano14020201_

Round 1

Reviewer 1 Report

Comments and Suggestions for Authors

Comments to the authorsï¼›

In this paper, the authors proposed a synaptic device with multilayer structure to reduce the operating power of neuromorphic computing systems while maintaining high-density integration. The topic is interesting, but the manuscript in its current form cannot be accepted and MAJOR REVISIONS are required.

1. The authors claim that the W/TaOX/AlOX/WOX stack was deposited on the side wall of the via-hole, but the schematic diagrams of devices shown in Fig.1 don’t reflect in this point, which may mislead readers that these are planar devices,

2. The cross-sectional TEM image in Fig.1 has the same problem. I suggest providing a cross-sectional image that reflects the overall view of the 3D devices.

3. From the EDS mapping, there seems to be W element in the region of TaOX, and Ta element in the region of WOX. Please explain this phenomenon.

4. I suggest to add IV curves of three devices under DC sweep mode, to intuitively reflect the resistive-switching characteristic.

5. In the triple layer device, how does the oxygen ions migrate from WOX to TaOX when there is a AlOX barrier layer (as shown in Fig2f)?

6. I think the difference between high and low resistance state is quite small, especially in the triple layer device, which means that the sense margin could be very small. Whether the signal can be correctly identified in an actual circuit network? And I notice that the weight values are significant overlapped in Fig2b and f, the accuracy of the network should be greatly affected.

7. The endurance and retention characteristics of devices should be demonstrated.

8. Some references about multilayer structured devices need to be added in the paper, such as:

1] 8-Layers 3D vertical RRAM with excellent scalability towards storage class memory applications. In 2017 IEEE International Electron Devices Meeting (IEDM) (pp. 2-7).

2] Memory switching and threshold switching in a 3D nanoscaled NbO x system. IEEE Electron Device Letters 40.5 (2019): 718-721.

Comments on the Quality of English Language

No comments.

Author Response

The authors would like to thank the reviewers for their timely and thorough review of the paper. The concerns of the reviewers have been addressed to present a more complete paper. We note that confusion has been caused by misrepresenting or using inappropriate language. These parts are clearly described in this decision letter and in the revised manuscript. The revisions to our manuscript are as the attached file.

Reviewer 2 Report

Comments and Suggestions for Authors

This paper proposed a synaptic device with a multilayer structure, to reduce the operating power of neuromorphic computing systems while maintaining high-density integration. The proposed synaptic device is experimentally validated and compared in the paper, and image recognition simulations were conducted.

1.    At present, there have been studies on various synaptic devices, and the work in this article should be compared with other existing works in terms of materials, power consumption, and working pressure.

2.    The description of image recognition technology is too simplistic, and relevant steps and experimental results should be provided.

3.    What is the connection method of the array, and a diagram should be provided.

4.    Is there a sneak-path current problem in the circuit?

5.    The reference discussion is not complete enough, and a review of other people's synaptic devices should be conducted.

Comments on the Quality of English Language

English language fine.

Author Response

(The authors gave the same response as above.)

Round 2

Reviewer 1 Report

Comments and Suggestions for Authors

Accept as it is.

Reviewer 2 Report

Comments and Suggestions for Authors

The author replied and resolved my issue, and the paper can be accepted.

Comments on the Quality of English Language

The English language is fine.